# Review of the Occurrence of Herbicides in Environmental Waters of Taihu Lake Basin and Its Potential Impact on Submerged Plants

**Yangyang Zhang, Min Hu and Aimin Li \***

State Key Laboratory of Pollution Control and Resource Reuse, School of Environment, Nanjing University, 163 Xianlin Avenue, Nanjing 210023, China; lovetadda@163.com (Y.Z.); 602022300062@smail.nju.edu.cn (M.H.)
\* Correspondence: liaimingroup@nju.edu.cn; Tel./Tax: +86-258-968-0508

**Abstract:** Over the past 20 years, a series of problems caused by eutrophication in Taihu Lake, and its surrounding environmental waters has received sustained and widespread attention. With the gradual extinction of submerged plants, which are the important basis for maintaining the aquatic ecological health of lakes, Taihu Lake has shifted from a grass-type clearwater lake to an algae-type turbid lake, posing severe challenges to the aquatic ecological health and security in this region. In addition, the occurrence of herbicides in the environmental waters of the Taihu Lake region has attracted the attention of several researchers. This study reviewed the evolution of submerged plants in Taihu Lake over recent decades. Moreover, the use of herbicides in the Taihu Lake region and their environmental occurrence in the past 20 years were statistically analyzed, and their toxic effects on submerged plants in previous reports summarized. Then, the potential impact of the environmental occurrence concentration of herbicides on submerged plants in the Taihu Lake region was evaluated. In conclusion, according to the results reported in the past paper, the environmental herbicide concentration in Taihu Lake has sometimes reached a level that can affect a variety of submerged plants, especially in the germination stage, which means that as an important cause of the degradation of submerged plants in shallow lakes, the effect of herbicides needs to be paid more attention to. The results of this review offer significant guidance for promoting science-based and standard use of herbicides and preventing their ecological risks in this region.

**Keywords:** aquatic environmental security; aquatic vegetation; ecological health; pesticides; shallow lake

## 1. Introduction

Aquatic ecosystems provide irreplaceable economic services to human society [1], but are currently experiencing severe losses compared with terrestrial ecosystems [2]. Lakes, especially shallow lakes, as the most important part of aquatic ecosystems, are facing a series of eco-environmental problems such as reduced water area, fragmented ecosystems, decreased biodiversity, and weakened ecological functions under human impacts [3–5]. The Taihu Lake has always been seen as a typical shallow lake in China. As the third largest freshwater lake in China, it serves as an important ecological function area in the Yangtze River Delta and a key ecological guarantee for the Yangtze River Economic Belt strategy [6]. Regrettably, since the blue-green algae bloom in Taihu Lake in 2007, it has suffered from water eutrophication for a long time [7].

Aquatic plants are the main primary producers of shallow lakes, playing a positive role in maintaining water transparency, accumulated biomass, and nutrient cycling. They are also the foundation for maintaining high biodiversity and stability of aquatic ecosystems [8,9]. Numerous reports have pointed out that aquatic vegetations, especially submerged plants, play a central role in the structures and functions of shallow lake ecosystems (Figure 1), and are the main producers of shallow lakes [2,10]. They affect a series of physical, chemical, and biological processes in lakes, and provide food, habitats, and

shelters for various aquatic organisms [11,12]. Unfortunately, the submerged plants in Taihu Lake are on the verge of disappearing [13–15]. However, the community succession of submerged plants in many shallow lakes in the Taihu Basin has occurred violently in recent decades, and submerged plants in some lakes have almost disappeared.

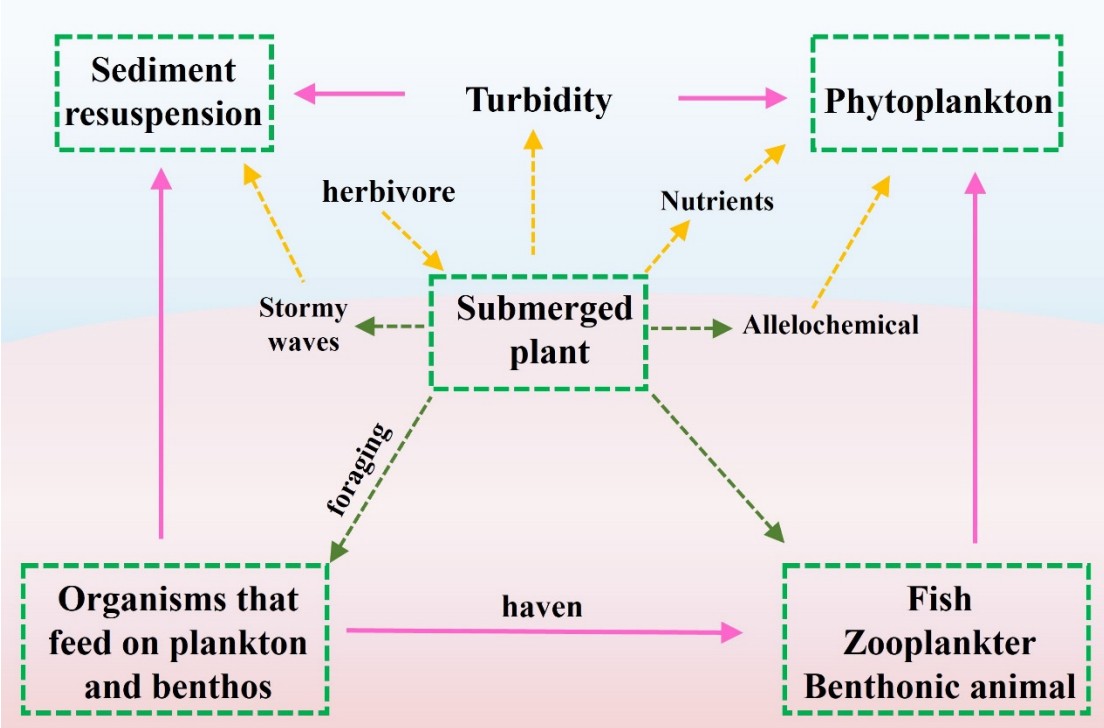

**Figure 1.** The role of submerged plants in ecosystems.

There are many environmental factors leading to a reduction in the distribution of submerged plants, such as water turbidity, water depth, sediment composition, water flow and waves, salinity, high eutrophication, dissolved oxygen level, and pesticides [16]. Additionally, biological interactions, such as competition with invasive plants, predation by a large number of herbivores [17], and inappropriate large-scale artificial fishing, also play a major role in the decline of submerged plants [18]. The disappearance of submerged plants in Taihu Lake and its surrounding large lakes has led to their transition from a state primarily dominated by macrophytes to a poor state dominated by phytoplankton and blue-green algae. Serious algal blooms occur frequently in summer, which has brought huge ecological threats and economic losses to the aquatic environment in this region [19].

Herbicides are one of the most widely used pesticides in the world. The use of herbicides on cultivated lands is still recognized as the most effective method to protect crops from weeds, but the surrounding environment is inevitably exposed to these chemicals [20,21]. The developed agricultural production activities in the Taihu Lake region annually consume a large amount of herbicides, which can pollute the nearby aquatic ecosystems through a variety of ways, including spray drift, runoff, precipitation, and leaching [22,23]. It was previously reported that even if the dose of herbicides is lower than 1% of the recommended dose, they will also have significant impacts on the growth, morphology, and reproduction of some non-target terrestrial plants [24], which are more sensitive to environmental stresses compared with terrestrial plants. Kemp et al. pointed out that exposure to herbicides is an important cause of degradation and biomass loss in submerged plant populations [25,26]. Consequently, among the threats faced by submerged plants in Taihu Lake, the exposure risk of herbicides may be non-negligible.

This study first reviewed the historical changes of dominant submerged plant populations in Taihu Lake in the past decades. Then, the local use and environmental occurrence of

herbicides over the past two decades were statistically analyzed. Based on the stress effects of herbicides on submerged plants reported in previous studies, whether the submerged plants are affected by the herbicide concentration in Taihu Lake was analyzed, and our views on whether herbicide may be an important factor in the disappearance or difficult regeneration of submerged plants in Taihu Lake are presented. The aim of this review was to offer a possible reason for the disappearance of submerged plants in Taihu Lake over the years, and to raise public awareness of the potential ecological impacts of herbicides, promoting the science-based and standard use of herbicides and preventing their ecological risks in this region.

## 2. Succession of Submerged Plants in the Taihu Lake

The Taihu Lake basin is rich in water resources, with developed water networks of numerous rivers and lakes. However, in the past decades, the aquatic environment of the Taihu Lake basin has been affected by eutrophication to varying degrees, and at the same time, submerged plants have also undergone profound changes [27,28].

### 2.1. Species

There are many types of submerged plants in Taihu Lake, including local wild species, artificial immigrant species, and alien invasive species [29]. As for living habits, submerged plants are immersed in water for a long time, and only their flower stalks and flowers come out of the water during flowering. Physiologically, the epidermal cells of submerged plants, usually without stratum corneum or wax layer, can directly absorb water, dissolved oxygen, and other nutrients [30]. Their roots degenerate or completely disappear. In addition, from the perspective of plant classification, although the genera, families, and species of submerged plants vary greatly, they all fall into the category of angiosperms and can be divided into monocotyledonous and dicotyledonous plants.

The investigation of submerged plants is based mainly on two methods: on-site visits and manual records, and satellite image recognition [31]. The former is high-cost and time-consuming, but the data obtained present high accuracy. The latter has high efficiency in data acquisition, but challenges in the fine recognition of submerged plant species [32,33]. From 2009 to 2010, An et al. [34] reported seven dominant submerged plants in Taihu Lake through an onsite investigation, including *Potamogeton maackianus*, *Hydrilla verticillata*, *Vallisneria spiralis*, *Zizania caduiftora*, *Elodea muttalli*, *Potamogeton malaianus*, and *Nymphoides peltata*. Using satellite image fine recognition technology, Gao et al. [35] identified 10 species of submerged plants, including curly pondweed, *Myriophyllum spicatum*, *Elodea canadensis*, *Vallisneria natans*, *Ceratophyllum demersum*, *Nymphoides peltatum*, *Potamogeton malaianus*, *Potamogeton pectinatus*, *Potamogeton maackianus*, and *Hydrilla verticillata*. Dong et al. and Zhang et al. [36,37] also reported results basically consistent with those of Gao et al. and An et al. Additionally, Li et al. [38] found that *Najas minor*, *Myriophyllum verticillatum*, and *Najas marina* are also the dominant species of submerged plants in Taihu Lake. According to the report of Wang et al. [39], *Elodea nuttallii* was artificially introduced in 1986, and *Cabomba caroliniana* is an alien invasive species identified in Taihu Lake in 2015. *Stuckenia pectinata* was introduced for unknown reasons, which may be that its distribution was not covered by the previous investigation, or it immigrated after water quality deterioration in Taihu Lake with its high pollution tolerance.

### 2.2. Spatiotemporal Distribution
2.2.1. Interannual Distribution

For a long time, the species, quantity, and distribution of submerged plants in Taihu Lake have been changing dynamically (Table 1). Based on the report of An Shuqing et al. [40], the distribution area of submerged vegetation in Taihu Lake was about 127.0 km$^2$ in 1981, which increased to 366.5 km$^2$ in 2005 (by 189%), and then suddenly fell to 163.3 km$^2$ in 2010. Zhang et al. [36] reported that in 1960, the distribution area of the submerged plant community, with *P. malaianus* + *V. natans* + *H. verticillata* as the main dominant

species, was about 160 km$^2$. By 1981, this data changed to 74 km$^2$, and approximately 7.3 km$^2$ *P. maackianus* community emerged. In 1996, the distribution area of submerged plant communities changed again; the *P. malaianus* + *V. natans* + *H. verticillata* community only had 7.4 km$^2$ left, while the *P. maackianus* community rapidly expanded to 51.7 km$^2$, and at the same time, approximately 14.7 km$^2$ *E. nuttallii* + *P. maackianus* community emerged. However, in 2002, the *P. maackianus* community completely disappeared, *P. malaianus* + *V. natans* + *H. verticillata* community restored to 12.2 km$^2$, and the *E. nuttallii* + *P. maackianus* community continued to climb to 22 km$^2$.

**Table 1.** Frequency and families of dominant species of submerged plants.

| Dominant Species | Families | Frequency | Years |
| --- | --- | --- | --- |
| *Potamogeton malaianus* | Potamogetonaceae | High | 1960–2014 |
| *Vallisneria natans* | Hydrocharitaceae | High | 1960, 1988, 2002, 2009–2010 |
| *Hydrilla verticillata* | Hydrocharitaceae | Low | 2002 |
| *Myriophyllum spicatum* | Halorrhagidaceae | Medium | 1988, 2009–2010 |
| *Potamogeton maackianus* | Potamogetonaceae | Low | 2014 |
| *Elodea muttalli* | Hydrocharitaceae | Medium | 1996, 2009–2010 |
| *Nymphoides peltata* | Menyanthaceae | Low | 2009–2010 |

In addition, Wang et al. [39] divided Taihu Lake into several regions for analysis (Figure 2), and also reported similar results. They stated that in 1960, only a small amount of *Potamogeton wrightii Morong* and *Vallisneria natans* was distributed on the periphery of *Phragmites australis* communities in the nearshore area of the Taihu Lake. By 1981, a large number of *Vallisneria natans* communities were found in the waters near Zhushan Bay at northern Taihu Lake and Yangwan ancient village in the east, while the distribution area of *Potamogeton wrightii Morong* communities reduced sharply, with the biomass only 11% of that in 1960. In 1988, *Vallisneria natans*, *Myriophyllum spicatum*, and *Potamogeton wrightii Morong* concentrated in the waters near Zhushan Bay at northern Taihu Lake and Yangwan ancient village in the east, becoming the dominant species. Among them, *Vallisneria natans* showed the highest dominance, *Myriophyllum spicatum* was distributed near Xuhu Lake of the eastern lake region, and *Potamogeton wrightii Morong* was scattered in different lake areas. In 1997, the aquatic vegetation in the northern lake region basically disappeared, while the northeastern, eastern, and southern lake regions were dominated by *Potamogeton wrightii Morong* and *Potamogeton maackianus*. In 2014, there was no aquatic vegetation distributed in the northern lake region; *Potamogeton wrightii Morong* communities in the eastern, northeastern, and southern lake regions showed an expanding trend from east to west, becoming the submerged plants with the highest dominance in the Taihu Lake. Moreover, it was reported that in 1987, the distribution of aquatic vegetation in the Taihu Lake was the closest to the state of nature, with large-area *Vallisneria natans* communities distributed in Meiliang Bay, Zhushan Bay, and Gonghu Bay. However, a survey in 1996 revealed that the aquatic vegetation in Meiliang Bay and Zhushan Bay almost disappeared completely, and subsequently, the cyanobacterial blooms in Taihu Lake gradually became rampant. In numerous later reports, this phenomenon is called the disappearance of aquatic vegetation in Taihu Lake caused by eutrophication.

In addition, the succession of several typical submerged plants was reported. From 1985 to 1995, the distribution area of *Vallisneria natans* in Taihu Lake dropped by 53.58 km$^2$, and mostly turned into water bodies and *Potamogeton maackianus*. Between 1995 and 2005, the distribution area of *Vallisneria natans* continued to fall by 17.94 km$^2$, mostly turning into water bodies and *Elodea nuttallii*. On the contrary, between 2005 and 2015, the distribution area of *Vallisneria natans* increased by 37.36 km$^2$, partially coming from natural water bodies and *Potamogeton maackianus*. After 2015, *Vallisneria natans* was close to extinction [41]. Over the 30 years from 1985 to 2015, the expansion of *Potamogeton wrightii Morong*, which can be distributed in areas with deep water and wind waves, was quite significant, mainly concentrated in the regions Dongshan Bay, East Taihu Bay, and Western Bay. In 1985,

*Potamogeton wrightii Morong* had already been a monodominant community in Dongshan Bay, with a net increase of 28.86 km$^2$ in 1985–1995, 49.89 km$^2$ in 1995–2005, and 6.67 km$^2$ in 2005–2015, mostly from water bodies. However, once a monodominant community was formed, *Potamogeton wrightii Morong* was unable to provide extensive and suitable living conditions for other aquatic organisms, resulting in a reduction in aquatic biodiversity and a serious impact on the ecological balance of lakes. *Elodea nuttallii* showed a net increase of 48.58 km$^2$ in 1985–1995, partially from *Potamogeton maackianus* and *Potamogeton wrightii Morong*, and of 43.32 km$^2$ in 1995–2005, partially from *Potamogeton maackianus*. As the net enclosure aquaculture in eastern Taihu Lake was basically saturated, the distribution area of *Elodea nuttallii* presented no increase from 2005 to 2015, and a stable community was formed in the region south of West Mountain Island. As for *Potamogeton crispus*, its distribution area decreased by 18.11 km$^2$ in 1985–1995, its net increase was 8.02 km$^2$ in 1995–2005, mainly from natural water bodies, and its net increase was 15.8 km$^2$ in 2005–2015, mainly distributed in Meiliang Bay, which is surrounded by mountains on three sides. Due to the relatively static environment, the peak period of algal blooms was avoided during the growing season, greatly promoting the prosperity of *Potamogeton crispus*. Moreover, from 1995 to 2005, the net increase of *Ceratophyllum demersum* was 49.11 km$^2$, partially from *Potamogeton maackianus* and natural water bodies. During 2005–2015, the net decrease of *Potamogeton maackianus* was 43.56 km$^2$, mostly turning into *Elodea nuttallii*. *Potamogeton maackianus* is a rootless submerged plant, distributed as a monodominant community in East Taihu Bay [39,41].

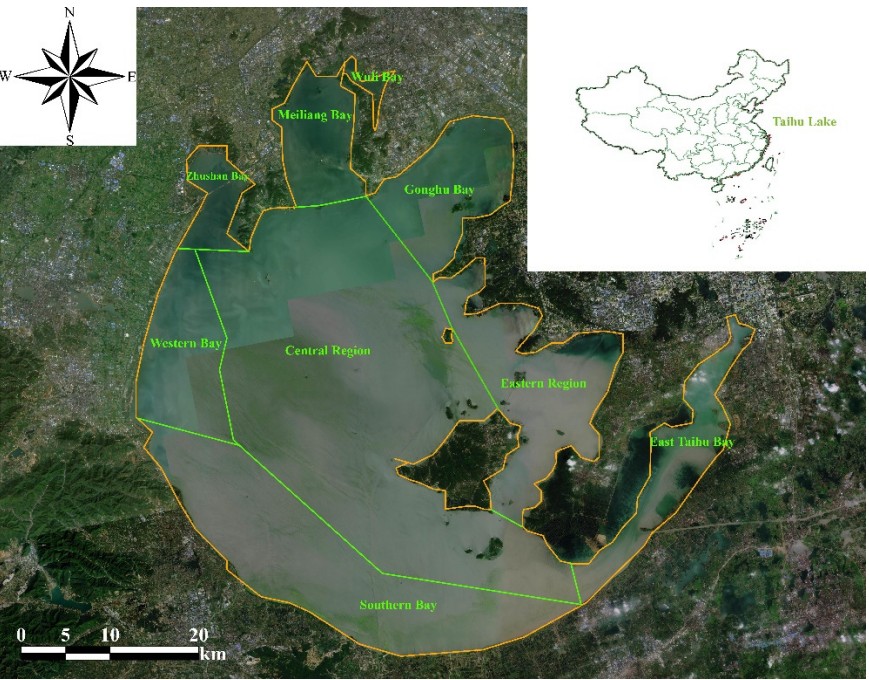

**Figure 2.** Location and partition of Taihu Lake.

### 2.2.2. Seasonal Distribution

In addition to the interannual differences, there are also significant seasonal differences in spatial distribution and biomass of the submerged plants in the Taihu Lake. Gao et al. [35] found that the inter-monthly variation trend of the distribution area of submerged plants in 2002 was the same as that in 2013. From January to March, there were no submerged plants in different regions of Taihu Lake. In April, a small amount of submerged plants began to appear in eastern, central, and southern bays, Gonghu Bay, and eastern Taihu Lake. From May to November, similar to the five regions mentioned above, submerged plants took up a large area. The distribution range reached its maximum in September. After September, it began to decline and approached zero in December. The inter-monthly variations of

submerged plant distribution can be roughly divided into three stages. At the first stage (January–March), the distribution area of submerged plants was quite small, only 14.57 km$^2$ in January 2002 and 3.65 km$^2$ in January 2013. At the second stage (April–September), the distribution area continuously increased from April, with the most significant increase from April to May. From May to September, it also continued to increase, but the velocity of increase was relatively low, such as 233.46 km$^2$ in September 2002 and 252.19 km$^2$ in September 2013. The third stage occurred from October to December, during which the distribution area began to decrease at an increasingly fast pace after September, and reduced to only 23.83 km$^2$ in December 2013. The regions with the largest distribution area were the eastern region (2002) and the eastern Taihu Lake (2013), 96.89 km$^2$ and 81.52 km$^2$, respectively. Winter is usually the dormant period for aquatic plants. Therefore, the distribution area is very small in winter, gradually increases in spring, and maintains a high level in summer and autumn.

The seasonal tendency of submerged vegetation is prominent in Taihu Lake. From the perspective of submerged plant species, *Vallisneria natans* mainly appears in summer or autumn, and is more common in waters with high transparency and hard substrate. The monodominant community of *Potamogeton maackianus* is only distributed in patches in the waters near Xuhu Bay and Gonghu Bay in summer. In addition, *Potamogeton wrightii Morong* is also the main dominant species in summer. In autumn, the dominance of *Myriophyllum spicatum* and *Potamogeton maackianus* rises, and a large number of communities dominated by *Myriophyllum spicatum* emerge, mainly concentrated in the waters with strong winds and high waves of Xuhu Bay. There are also many *Myriophyllum spicatum* distributed around the net enclosure of eastern Taihu Lake. In winter, *Potamogeton crispus* communities are widely distributed in still water areas, which may be related to the fact that it is the unique biennial aquatic plant in Taihu Lake. Generally, it begins to germinate in November, and can even become an absolute dominant species in local lake areas in spring [39].

## 3. Environmental Source and Occurrence of Herbicides in Taihu Lake

Herbicides are a class of artificially synthesized chemicals used for complete or selective withering of the target weeds. At present, weed control in China is still mainly based on the application of chemical herbicides, with their registered quantity, production, sales, and use accounting for over one third of pesticide products [42].

### 3.1. Classification of Herbicides

Herbicides can be classified into different types based on their usage, conductivity, mode of action, and chemical structure. The functions and environmental risks of herbicides are closely related to their chemical structures, and herbicides with similar structures often have common targets and weeding mechanisms [43,44]. Therefore, this study provided statistics of common herbicides based on their chemical structures (Table 2).

The weeding mechanism of herbicides is mainly to disrupt the normal signal transmission and metabolic pathways of target plants. For example, carbamates and sulfonylureas severely inhibit the activity of acetyl lactate synthase (ALS), leading to valine and isoleucine deficiency, which can stop plant cell mitosis at the gap 1 (G1) and gap 2 (G2), and cause growth arrest and ultimate death of weeds [45]. Ureas and triazines mainly act on the D1 protein of photosystem II (PSII), inhibiting plant photosynthesis [46]. Amides are generally lipid synthesis inhibitors or cell division inhibitors [47]. Phenoxyalkanoic acids are all developed based on the structure of 2,4-D as hormone herbicides. The symptoms of plant poisoning are similar to those acted by auxin [48]. The mechanisms of pyridines vary with the specific type of herbicides, including typical auxin effects, inhibition of carotene biosynthesis, cell division inhibitors, and inhibition of polar auxin transport [49].

**Table 2.** Classification of common herbicides based on chemical structures.

| No. | Type of Herbicide | Representative Compounds |
|-----|-------------------|--------------------------|
| 1 | Triketones | Mesotrione, mesotrione, etc. |
| 2 | Pyrazoles | Cypyrafluone, bipyrazone, tripyrasulfone, fenpyrazone, etc. |
| 3 | Pyridines | Fluroxypyr, halauxifen-methyl, etc. |
| 4 | Sulfonylureas | Tribenuron, bensulfuron-methyl, nicosulfuron, pyrazosulfuron-ethyl, mesosulfuron-methyl, etc. |
| 5 | Sulfonamides | Florasulam, penoxsulam, pyroxsulam, flumetsulam, etc. |
| 6 | Pyrimidylsalicylates | Isopropyl ether, pyribenzoxim, bispyribac-sodium, pyrithiobac-sodium, pyriftalid, etc. |
| 7 | Imidazolinones | Imazethapyr, imazamox, etc. |
| 8 | Aryloxyphenoxypropionates | Fenoxaprop-p-ethyl, quizalofop-p-ethyl, clodinafop-propargyl, cyhalofop-butyl, metamifop, etc. |
| 9 | Phenoxyalkanoic acids | 2,4-D, 2-methyl-4-chlorophenoxyacetic acid, clodinafop-propargyl, clopyralid, fluazifop-p-butyl, haloxyfop-P-methyl, fenoxaprop-p-ethyl, etc. |
| 10 | Triazines | Atrazine, desmetryn, prometryne, terbuthylazine, etc. |
| 11 | Amides | Pretilachlor, acetochlor, metolachlor, butachlor, etc. |
| 12 | Dinitroanilines | Trifluralin, pendimethalin, butralin, etc. |
| 13 | Cyclohexanediones | Sethoxydim, clethodim, etc. |
| 14 | Ureas | Isoproturon, chlortoluron, diuron, etc. |
| 15 | Diphenylethers | Fomesafen, fluoroglycofen-ethyl, oxyfluorfen, etc. |
| 16 | Cyclic imines | Oxadiazon, flumiclorac-pentyl, etc. |
| 17 | Carbamates | Thiobencarb, phenmedipham, molinate, etc. |
| 18 | Organophosphates | Glyphosate, glufosinate ammonium, etc. |
| 19 | Bipyridines | Paraquat, diquat dibromide, etc. |
| 20 | Heterocycles/Others | Bentazone, etc. |

*3.2. Targets of Herbicides*

The Taihu Lake region is a major grain-producing area in China, with a highly developed agricultural planting industry. Grain and oil crops including rice, wheat, oil-seed rape, corn, and soybean, are the main agricultural products in this region [50]. Additionally, the cultivation and maintenance of fruits, vegetables, and tea also consume a large amount of herbicides. Moreover, the weeds on ditches, roads, and other non-agricultural or idle lands are sometimes cleared up with herbicides [51]. Table 3 summarizes the herbicides commonly used in agricultural production in Taihu Lake area.

The use of herbicides is closely related to the growth cycle of crops. Taking grain and oil crops as an example, the growth period of rice is around May–October each year [52]. According to traditional transplanting methods, the peak period for herbicide use is the turning-green stage of rice after transplanting, early June each year [53]. The growth period of wheat is approximately from late October to early June of the next year. The peak period for herbicide application is divided into the seedling stage (mid–late November) and the regeneration stage (late February–early March of the next year). Considering the effect of weed control and its impact on the next round of crops, the former generally occupies a dominant position [54]. The growth cycle and time nodes for weed control of oil-seed rape are similar to those of winter wheat [55]. As for corn, the growth period is around late May to early October each year, and the peak period of herbicide application is from June to early July. The sowing of soybeans lags slightly behind that of corn, with a growth period

from mid-June to late September, and a peak period for herbicide application from late June to early July [56,57].

**Table 3.** Herbicides commonly used in grain and oil crops.

| Crop | Growth Period | Types of Applied Herbicides |
| --- | --- | --- |
| Rice | May–October | Chlorimuron-ethyl, pyrazosulfuron-ethyl, bensulfuron-methyl, penoxsulam, bispyribac-sodium, cyhalofop-butyl, 2,4-D, 2-methyl-4-chlorophenoxyacetic acid, prometryne, butachlor, pretilachlor, pendimethalin, oxadiazon, bentazone, quinclorac |
| Winter wheat | Early October–early June of the next year | Carfentrazone-ethyl, fluroxypyr, thifensulfuron-methyl, mesosulfuron-methyl, flucarbazone-sodium, florasulam, pyroxsulam, clodinafop-propargyl, fenoxaprop-p-ethyl, isoproturon, 2,4-D butyl ester, pinoxaden |
| Oil-seed rape | September to November-May of the next year | Acetochlor, butralin, trifluralin, clopyralid, fluazifop-p-butyl, haloxyfop-p-methyl, benzoic acid, quizalofop-p-ethyl, fenoxaprop-p-ethyl, benazolin |
| Corn | June–October | Acetochlor, metolachlor, nicosulfuron, atrazine, mesotrione, 2,4-D, diquat dibromide, glyphosate, glufosinate-ammonium |
| Soybean | Mid-June–late September | Quizalofop-p-ethyl, haloxyfop-p-methyl, 2,4-D butyl ester, thifensulfuron-methyl, clethodim, fluoroglycofen-ethyl, fomesafen, butralin, acetochlor, prometryne, quizalofop-p-ethyl, bentazone |

Improper application of herbicides will lead to crop yield reduction or even crop failure. Butachlor is always used in rice fields to kill annual Poaceae weeds and certain broad-leaved weeds, while bentazone and 2,4-D are applied to kill annual broad-leaved weeds and sedges. However, they are ineffective against Poaceae weeds. Quinclorac has a special effect in killing barnyard grasses in rice fields [58,59]. Usually, herbicides are only applied once per crop of rice. As for winter wheat, carfentrazone-ethyl, fluroxypyr, florasulam, and thifensulfuron-methyl are generally used to control broad-leaved weeds, while herbicide products containing effective ingredients such as mesosulfuron-methyl, fenoxaprop-p-ethyl, flucarbazone-sodium, pinoxaden, pyroxsulam, and clodinafop-propargyl are selected for Poaceae weeds according to the grass phenophase, which are mainly applied before the turn of the year [60]. Benazolin and clopyralid are commonly used in oil-seed rape fields to kill broad-leaved weeds, while quizalofop-p-ethyl, fluazifop-p-butyl, haloxyfop-p-methyl, and fenoxaprop-p-ethyl are applied to kill Poaceae weeds. In corn fields, amides, triazines, phenoxyalkanoic acids, and sulfonylureas are mainly used [61]. Among them, amides are currently the most widely used herbicides in corn fields, which can effectively control annual Poaceae weeds and some annual broad-leaved weeds. Phenoxyalkanoic acids and sulfonylureas are mainly applied for the control of broad-leaved weeds from corn seedlings [62]. Common herbicides for controlling broad-leaved weeds in soybean fields include bentazone, fluoroglycofen-ethyl, and acifluorfen, while those for controlling Poaceae weeds include phenoxyalkanoic acids (such as quizalofop-p-ethyl) and cyclohexenones (such as clethodim). Moreover, imazethapyr and fomesafen are applied to control both Poaceae and broad-leaved weeds [63].

For other agricultural productions, such as tuber crops (sweet potatoes, potatoes, etc.), trifluralin, acetochlor, and metolachlor are mainly used [64]. Cotton is generally treated with herbicides including trifluralin, pendimethalin, prometryn, acetochlor, and oxyfluorfen. Occasionally, sterilant herbicides such as glyphosate and glufosinate-ammonium are also selected to remove unearthed weeds before crop planting [65]. Phenoxyalkanoic acid 2,4-D is the most common herbicide specific to *Cyperus rotundus* in sugarcane fields. Weeds in

tea fields can be divided into two categories of spring weeds and summer weeds. When there are overgrown spring and summer weeds, imazethapyr and fomesafen which are herbicides for both Poaceae and broad-leaved weeds, can be sprayed once in mid-May and late July, respectively [66]. In addition, for arboreous or vine fruit trees in orchards, sterilant herbicides can be selected to completely remove weeds [67].

It is particularly important to note that the selection and application of herbicides must strictly follow science laws. The meteorological conditions during application, soil moisture, and crop and weed status are the main bases for herbicide selection. Additionally, continuous use of a single type of herbicide should be avoided to reduce weed resistance, which can weaken the efficiency of herbicides [68]. Therefore, based on the crop types in the survey area, the types of herbicides present in the environment can only be determined roughly, and on-site visits and non-targeted screening are needed to obtain more detailed information for further verification.

### 3.3. Environmental Occurrence of Herbicides

The identification, fate, and spatiotemporal load of pesticides at the watershed or regional scale are important components for achieving aquatic ecological security and sustainable agriculture. At present, pesticides and organochlorine pesticides are the main subjects for the occurrence of pesticides in the Taihu Lake region [69,70]. However, there are few surveys on the environmental occurrence concentration of herbicides in water bodies, which may be related to the incomplete content of non-target aquatic plants in the current ecological risk assessment of environmental chemicals.

The herbicides in the environmental waters of the Taihu Lake region mainly come from the loss during their use. Figure 3 shows the process of herbicide migration from the rice field to the important environmental water body. The loss of herbicides in the field and the process of entering natural water bodies are mainly jointly controlled by their physicochemical properties, cultivation methods in farmlands (drylands and paddy fields), and hydraulic and other natural conditions in the usage area. Among them, the physicochemical properties of herbicides (such as octanol–water distribution coefficient and half-life) are key factors affecting their fates. For example, the half-life of metolachlor in water is 88 days [71]. Acetochlor has high stability and persistence in soil, with a half-life of 6.51–26.7 days, which varies with the changes in environmental conditions [72,73]. The half-life of prometryne in aquatic environments is 56 days. The half-life of 2,4-D ranges from 20–200 days, and it has high water solubility and does not tend to transform into sediments [74]. The half-life of atrazine is between 1–12 months [75]. Although the theoretical half-life of tribenuron is only about 8–10 days, its residual period (the time required for content reduction in soil by more than 75%) is about 60 days [76]. Therefore, high environmental persistence indicates that herbicides may persist in aquatic environments (including sediment) for a long time and pose a risk to non-target aquatic plants in the environments.

Sun et al. carried out statistics on the environmental concentration of atrazine at 10 sites in Meiliang Bay of Taihu Lake from November 2003 to August 2004. The concentration range of atrazine was 25.8–614 ng/L (average, 251.5 ng/L) in November 2003, 128–506 ng/L (average, 277.1 ng/L) in January 2004, and 21.6–260 ng/L (average, 132.3 ng/L) in August 2004. It is speculated that it may mainly come from the surrounding winter wheat and oil-seed rape fields [77]. Xu et al. surveyed herbicides at five sampling sites in the Taihu Lake region in July 2014, revealing that the concentration of atrazine averaged 352.4 ng/L, and reached 600–750 ng/L at some sites. However, the average concentrations of prometryne and simazine were only 42.6 ng/L and 41.2 ng/L, much lower compared with atrazine [78]. Zhang et al. investigated the Qinhuai River, the Yangtze River, and Taihu Lake from April to May 2016, respectively, and found that the six herbicides tested were detected at all sites. The maximum environmental concentrations of isoproturon, atrazine, 2-hydroxyatrazine, and terbutryn were 847 ng/L, 1726 ng/L, 2680 ng/L, and 1687 ng/L, respectively, while the maximum concentrations of metolachlor and diuron were only 316 ng/L and 107.3 ng/L [79]. Xian et al. measured the environmental concen-

trations of nitrapyrin, atrazine, acetochlor, and metolachlor in the Jiuli River, the main river entering Taihu Lake, in the spring, autumn, and winter of 2018, respectively. Among the six sampling sites, the detection rates of atrazine, metolachlor, nitrapyrin, and acetochlor were 100%, 72.2%, 22.2%, and 13.9%, with the concentration range of 19.1–1190 ng/L, −94.3 ng/L (undetected), −25.7 ng/L (undetected), and −26 ng/L (undetected), respectively. Combining the results of Xu and Zhang et al., it largely reflects that atrazine has a long history of use and is common in the Taihu Lake region. Moreover, they found that the detection rate of herbicides at each sampling site was higher in May than in other months, while that in October was significantly lower than in other months. It is speculated that May may be the peak of weeding in rice fields of this region, while October happens to be a blank period when the previous stage of agricultural activities ends and the next stage has not yet begun, so the concentration of herbicides is relatively low [80].

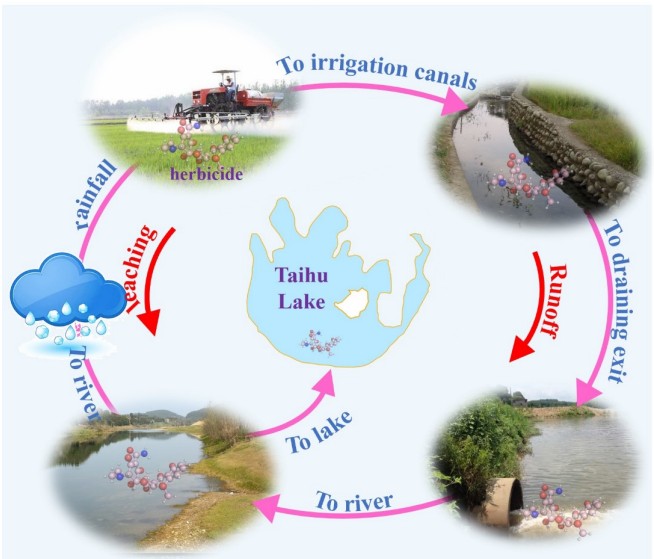

**Figure 3.** Schematic diagram of main sources and access routes of environmental herbicides in Taihu Lake area.

In addition, Dai et al. detected sulfonylureas in the aquatic environment at 33 sampling sites set at 22 main rivers entering the Taihu Lake and the Taihu Lake in October 2019, which revealed a detection rate of 100%, with a range of total concentration of 1.69–38.1 ng/L (average concentration, 19.4 ng/L). The detection rate of sulfonylureas in 22 rivers entering the lake was 90.9%, with a total concentration of ND −70.7 ng/L, indicating a relatively low total concentration [81].

Recently, Yu et al. reported in detail the environmental occurrence of herbicides in the Taihu Lake region in four seasons [82]. It was discovered that in March, the total concentration of herbicides in the surrounding areas of Taihu Lake was relatively high, which is speculated to be caused jointly by agricultural activities in winter and early spring and low water flow. The intensive agricultural activities in late summer, combined with the gradual decrease in precipitation, led to the peak of pesticide concentration in September. In March, the total herbicide concentration near the entrance of the Grand Canal was relatively high, mainly 2,4-D (1590 ng/L, accounting for 92% of the total herbicide concentration at this sampling site). At most sampling sites, 2,4-D, bentazone, isoproturon, and pretilachlor were often detected, accounting for over 50% of the total herbicide contamination. In June, the herbicide in the environment was mainly 2,4-D, and the sites with higher concentrations were mainly distributed near Changdang Lake and Gehu Lake in the upstream of Taihu Lake, with the maximum total herbicide concentration reaching 1530 ng/L. The herbicides with the highest concentration in September were bentazone (average concentration, 234 ng/L; detection frequency, 100%), metolachlor (126 ng/L, 100%), sulfonylurea nicosul-

furon (834 ng/L), and tribenuron-methyl (155 ng/L). In December, the herbicide with the highest concentration was atrazine, but its average concentration was only 15 ng/L. Based on the above reports, Table 4 summarizes several herbicides with high environmental detection frequency in Taihu Lake and their potential sources.

**Table 4.** Herbicides with high frequency and their types that were detected in Taihu Lake area.

| Type | Species | Frequency | Potential Source |
|---|---|---|---|
| Phenoxy carboxylic acid | 2,4-D | High | Rice, wheat |
| Triazines | Atrazine | High | Maize |
| Ureas | Isoproturon | Medium | Rice, orchard |
| Sonylurea | Tribenuron-methyl, sulfonylurea nicosulfuron | High | Rice |
| Amides | Metolachlor, pretilachlor | High | Rice, wheat |
| Others | Bentazone | Medium | Rice, wheat |

## 4. Impact of Herbicides on Submerged Plants

The widespread use and frequent detection of herbicides in the environment have raised concerns among researchers about their potential environmental impacts. It has been demonstrated that even if the dose is less than 1% of the recommended dose, it will have significant impacts on the growth, morphology, and reproduction of some non-target terrestrial plants [24], while aquatic ecosystems may be more sensitive to herbicides than terrestrial ecosystems [83]. However, most current studies evaluating the risk of herbicides to aquatic environments are conducted through experiments on various model animals (zoobenthos, water fleas, and planktonic microorganisms), floating-leaved plants and algae [84,85], while research on the impact of herbicides on submerged plants has been rarely seen.

There are many factors that affect the toxic effects of herbicides on submerged plants. In addition to the internal factors of herbicides and submerged plants, external factors, including herbicide concentrations, the coexistence of multiple herbicides, the half-life of herbicides in aquatic environments, complex interference conditions in aquatic environments (turbidity, natural organic matters, various nutrient salt concentrations, etc.), and other biological factors, can all interfere with the toxic effects of herbicides on submerged plants. There are also many indicators that reflect the toxic effects of herbicides on submerged plants, including plant biomass, specific leaf area, and relative growth rate macroscopically, as well as photosynthetic pigment content, organic osmoregulatory substances, reactive oxygen species (ROS) level, a series of stress-resistant enzyme activities, and plant auxin level physiologically and ecologically (Figure 4).

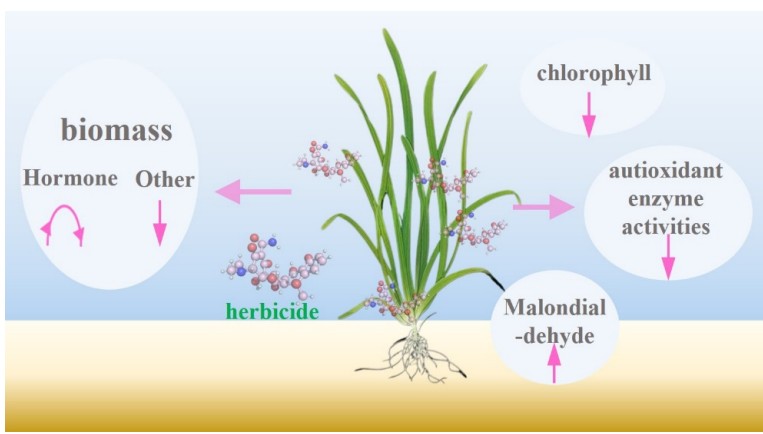

**Figure 4.** Diagram of stress response of submerged plants under herbicide stress.

### 4.1. Apparent Growth Status

The apparent growth status of plants is a comprehensive reflection of their health. For submerged plants, a healthy growth state enables them to maintain and develop biomass, absorb nutrient salts from water bodies, reduce eutrophication, and play other ecological roles in aquatic environments.

Gao et al. [86] studied the toxic effects of four types of herbicides, including butachlor, quinclorac, bensulfuron-methyl, and atrazine, on submerged plants such as *Ceratophyllum demersum*, *Vallisneria natans*, and *Elodea nuttallii*. The results showed that quinclorac can cause curly leaves and slender stems of *Ceratophyllum demersum*, and, the same as bensulfuron-methyl, it can lead to etiolation to varying degrees in *Ceratophyllum demersum*. In addition, bensulfuron-methyl can reduce the root vitality of *Vallisneria natans*, and, like butachlor, it can result in leaf wilting and a significant reduction in tillering of *Elodea nuttallii*. Butachlor, even at a concentration of 0.1 μg/L, can also significantly inhibit the growth of *Ceratophyllum demersum* and *Elodea nuttallii*. Reaching a concentration of 0.5 μg/L, it can cause death in all *Vallisneria natans.* At a concentration > 5 μg/L, it can lead to the complete death of *Elodea nuttallii* within a week.

Relative growth rate (RGR) is an important indicator for measuring the growth rate of plants whose population density mainly depends on vegetative reproduction. However, toxic stress may sometimes take several days or even weeks to affect growth rate, so ALS can better display the effects on plant cell division at a short time scale [87].The study of Cedergreen et al. demonstrated that metsulfuron has significant stress effects on different submerged plants [88]. When evaluating toxicity based on RGR, only a significant dose-response relationship was found between *Ceratophyllum demersum* and metsulfuron concentration, with an $EC_{50}$ value of 4.13 ± 3.32 μg/L. When evaluating toxicity with ALS, the $EC_{50}$ value was 0.57 ± 0.26 μg/L in *Elodea nuttallii*, 0.2 ± 0.12 μg/L in *Ceratophyllum demersum*, 0.29 ± 0.31 μg/L in *Myriophyllum verticillatum*, 0.23 ± 0.23 μg/L in *Potamogeton crispus*, and even as low as 0.07 ± 0.04 μg/L in *Batrachium bungei*. This finding is consistent with the consensus that metsulfuron has a rapid inhibitory effect on cell division but no significant effect on photosynthesis. In addition to metsulfuron, Chang et al. believe that submerged plants are all quite sensitive to most sulfonylureas. Their study revealed that application with 3.3 μg/L sulfosulfuron for 70 days can reduce the biomass of *Myriophyllum verticillatum* by about 48% [89]. Gao et al. also found that both *Elodea nuttallii* and *Ceratophyllum demersum* were sensitive to bensulfuron-methyl, the former being more sensitive [86]. Bensulfuron-methyl at 0.5 μg/L can severely restrict the RGR of *Elodea nuttallii*. On the contrary, quinclorac at 5–10 μg/L can increase the biomass of submerged plants in that it is an auxinic herbicide, and its low concentrations can promote plant growth.

Richardson et al. [90] reported changes in the dry matter of submerged plants in a series of simulated halauxifen-methyl aqueous solutions. The $EC_{50}$ values were 0.11 μg/L and 0.23 μg/L in *Myriophyllum verticillatum*, 1.4 μg/L and 2.5 μg/L in *Hydrilla verticillata*, and 6.9 μg/L and 13.1 μg/L in *Elodea nuttallii* on 14 days and 28 days, respectively. In real aquatic environment experiments, the $EC_{50}$ values of *Myriophyllum verticillatum* on the 14 and 28 days were 0.12 μg/L and 0.58 μg/L, respectively, without significant changes. Netherland et al. showed similar experimental results in that the $EC_{50}$ values for fresh and dry weight of *Hydrilla verticillata* were 0.94 μg/L and 0.71 μg/L, respectively, and those of *Myriophyllum verticillatum* were both 0.3 μg/L. They also found that the toxicity of halauxifen-methyl decreased in acidic environments, and its $EC_{50}$ value even increased by 100 times in *Myriophyllum verticillatum* [91].

Additionally, it has been reported that atrazine also has a significant impact on submerged plants. In ponds, atrazine at as low as 20 μg/L can reduce the biomass of *Myriophyllum verticillatum* by 60%, and long-term exposure can lead to its complete death. Under this concentration, *Potamogeton* will be replaced by other aquatic plants, and at a concentration of 50 μg/L, another submerged plant, *Najas marina*, will also be replaced by other aquatic plants [92]. Atrazine mainly acts on the pathways of plant photosynthesis. When plants

are exposed to atrazine for a short time, they may recover from the toxic effects of atrazine. Another report pointed out that the atrazine concentration affecting freshwater phytoplankton reaches as low as 1 μg/L, and that affecting saltwater macrophytes reaches 5 μg/L. The EC$_{50}$ value for the root RGR of *Hydrilla verticillata* on 4 days was about 430 μg/L, and that for the leaf RGR on 28 days was only 80 ug/L. An amount of 53 ug/L was required for 21-day mortality of 100% for *Potamogeton distinctus*. Unlike other studies, they found that atrazine had an EC$_{50}$ value as high as 1104 ug/L on the leaf RGR of *Myriophyllum verticillatum* on 28 days [93]. Moreover, there are reports that, over time, plants may adapt to atrazine stress by increasing chlorophyll content, such as compensating for reduced photosynthesis by increasing chlorophyll content, especially at low to moderate exposure concentrations; this adaptation may be important [94].

What is more, 2,4-D is a widely existing herbicide in the environmental waters of Taihu Lake. Ryan M. Wersal et al. believed that only 0.25 mg/L 2,4-D can produce 85% control on *Myriophyllum verticillatum* within 72 h in the wild [95]. Zhang et al. found that 40–100 μg/L 2,4-D had severe effects on the roots, branches, and new structures of *Myriophyllum verticillatum*, and at a concentration as low as 20 μg/L, it can also cause stipe swelling and tissue yellowing. When the concentration reaches >200 μg/L, *Myriophyllum verticillatum* would die within 5 days [96]. However, according to the US EPA/OPP aquatic life benchmark, the environmental concentration limits for herbicides protecting non-vascular plants and vascular plants are 1.64 μg/L and 2.3 μg/L, respectively, which means that when multiple herbicides coexist in the environment, their total concentration should not exceed these limits. Moreover, due to the fact that most submerged plants are in the germination period from March to April, they are more sensitive to herbicide exposure than adult plants [97]. This period coincides with the season of secondary use of herbicides for wheat and oil-seed rape in spring, so the probability of damage to submerged plants will rise.

*4.2. Physiological and Ecological Indicators*

Due to structural and functional differences, the impacts of different types of herbicides on the physiological and ecological indicators of submerged plants vary significantly. The photosynthetic pigments of higher plants are the material basis for photosynthesis, mainly including chlorophyll a (Chl-a), chlorophyll b (Chl-b), and carotenoids (Car). Chl content can reflect the growth status and photosynthetic capacity of plants. Car is an important photosynthetic pigment in aquatic plants, which acts as an auxiliary light-harvesting pigment in photosynthesis, protecting Chl from photooxidative damage. Therefore, abnormal reduction in Car content may lead to plant bleaching, oxidative damage, and even death.

According to the research results of Gao et al., both 0.5 μg/L butachlor and quinclorac can seriously affect the Chl-a content of *Ceratophyllum demersum* in a short period of time, which cannot be recovered in the following 21 days. In addition, 5 μg/L atrazine and 50 μg/L bensulfuron-methyl are required to effectively inhibit the Chl content of *Ceratophyllum demersum*. For *Vallisneria natans*, 0.5 μg/L butachlor, bensulfuron-methyl, and atrazine can all cause significant Chl-a loss in a short period of time [86]. The most sensitive submerged plant is *Elodea nuttallii*, and only 0.1 μg/L butachlor or 0.5 μg/L bensulfuron-methyl can cause almost complete loss of its Chl content. The research results of Wu et al. show that the pyrrolidinone herbicide flurochloridone also has a significant impact on the photosynthetic pigments of *Ceratophyllum demersum*. An amount of 20 μg/L flurochloridone can reduce the contents of Chl-a, Chl-b, and Car in *Ceratophyllum demersum* by half, but the inhibitory effect of flurochloridone with increasing dose on Chl is not significant [83]. Moreover, Ma et al. conducted extensive research on the impacts of atrazine on macrophytes and algae, and found that its EC$_{50}$ value for photosynthesis of submerged plants ranged from 22 to approximately 474 mg/L, with only a few exceptions. Therefore, in most cases, atrazine does not have a significant impact on the photosynthetic system of submerged plants.

In addition, ROS, including superoxide anions ($O_2{}^-$), hydrogen peroxide ($H_2O_2$), and hydroxyl radicals (OH), can act as signaling molecules to regulate plant development and initiate plant responses to environmental conditions at low concentrations. However, excessive ROS produced by plants under environmental stresses can oxidize biofilms, and alter the fluidity and permeability of cell membranes, resulting in toxic effects on plants. Meanwhile, plants have antioxidant enzyme systems, including superoxide dismutase (SOD), catalase (CAT), and peroxidase (POD), which can remove excess ROS and protect cells from membrane–lipid peroxidation. The content of the final product of membrane–lipid peroxidation, malondialdehyde (MDA), directly reflects the level of oxidative damage to plants by ROS [98]. Moreover, soluble proteins are important osmoregulatory substances in plants, and also an important indicator for evaluating plant responses to stresses, serving as the important material basis for the structure and function of plant organs.

According to the research results of Wu et al., with the increase in flurochloridone concentration, the SOD activity in *Ceratophyllum demersum* showed a decreasing trend. Flurochloridone above 1000 μg/L significantly affected the MDA content of *Ceratophyllum demersum*, indicating oxidative damage to the membrane. Additionally, soluble proteins decreased with the increase in flurochloridone dose. When the concentration of flurochloridone exceeded 100 μg/L, SOD activity significantly decreased, due to the irreversible damage caused by excessive free radicals to antioxidant enzymes. Moreover, when flurochloridone concentration exceeded 300 μg/L, as with the accumulation of ROS, plant cells were irreversibly damaged, leading to a significant drop in protein and soluble sugar contents [83].

## 5. Discussion

According to the contents of Chapter 4, numerous herbicides have strong toxic effects on common submerged plants in Taihu Lake area, namely, 2,4-D (phenoxycarboxylic acids), butachlor (amides), bensulfuron-methyl (sulfonylurea), and others (like bentazone). It is to be expected that most herbicides classified according to their chemical structure will have similar effects on the same type of submerged plants. At present, although no herbicide type with an effect concentration higher than half of the growth rate of submerged plants has been detected in the water environment of Taihu Lake region, it should be noted that the toxicity data of herbicides discussed above on submerged plants are based on the results of laboratory simulation studies, and submerged plants are obviously more vulnerable when facing more complex living conditions in the real environment. At the same time, there are many kinds of herbicides in the real environment, which can destroy the physiological functions of submerged plants through various modes of action, which will increase the risk of toxicity of submerged plants. In addition, according to the content of Chapter 2, we found that the germination stage of submerged plants coincided with the peak of herbicide use in Taihu Lake, which means that a short period of high environmental concentration of herbicides would cause irreversible damage to submerged plants. Due to the lack of environmental research, the relatively high concentrations of environmental herbicides were not monitored, so this risk could easily be ignored.

In the coming period, to protect the ecological health of the water environment, starting from the recovery of submerged plants, policy managers in the Taihu Lake region will need to conduct a great deal of effort, which must include stricter control of herbicides discharged into the water environment, and actively collect and track more detailed data on the spatial and temporal distribution of herbicides in the local water environment. Of course, this effort also needs to be supported and coordinated by local agricultural practitioners, and it is important to note that all actions need to be carried out without disrupting agricultural production, which may require closer international cooperation, such as the development of more effective and less toxic herbicides.

## 6. Conclusions

At present, Taihu Lake and its surrounding shallow lakes, to a certain degree, are still facing eutrophication. Although the latest data show that the environmental water quality has improved from ten years ago, the problem of submerged plants being damaged and difficult to recover naturally in the short term still exists. It is worth noting that the agricultural planting industry is highly developed, the intensity of pesticide use is high, and the environmental occurrence concentration is high in the Taihu Lake region. Compared with traditional pollutants such as nitrogen and phosphorus, novel herbicides from agricultural sources also pose a great risk to the restoration of submerged plant communities in Taihu Lake. Therefore, it is considered necessary to control and reduce the risk of herbicide toxicity to submerged plants in the Taihu Lake region. There are likely to be three urgent tasks in the future. First, a systematic environmental herbicide concentration survey should be established in the Taihu Lake region, including setting fixed monitoring sites and increasing monitoring intensity at key time nodes, such as the peak period of herbicide use for major crops, including wheat and rice. Second, herbicides should be prevented from entering natural water bodies from the source. The characteristics of local farmlands should be fully utilized, and a backflow of irrigation water in the fields should be formed to reduce the discharge of herbicides, especially during the germination period of submerged plants. Third, artificial planting of submerged vegetation should be conducted in major water bodies to restore and develop the biomass of submerged plants. The water should be further purified while enhancing its resistance to environmental micropollutants by utilizing the absorption and degradation capabilities of submerged vegetation.

**Author Contributions:** Conceptualization, Y.Z. and M.H.; methodology, A.L.; software, Y.Z.; validation, Y.Z. and M.H.; formal analysis, Y.Z.; investigation, Y.Z.; resources, Y.Z.; data curation, Y.Z.; writing—original draft preparation, Y.Z.; writing—review and editing, Y.Z.; visualization, Y.Z.; supervision, Y.Z.; project administration, A.L.; funding acquisition, M.H. All authors have read and agreed to the published version of the manuscript.

**Funding:** This paper was financially supported by the National Natural Science Foundation of China (No. 52270072, 42227806), The National Key Research and Development Program of China (No. 2023YFE0100900).

**Conflicts of Interest:** The authors declare no conflict of interest.

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
