# Peer review of "Review of the Occurrence of Herbicides in Environmental Waters of Taihu Lake Basin and Its Potential Impact on Submerged Plants"

_water, doi:10.3390/w16050726_

Round 1

Reviewer 1 Report

Comments and Suggestions for Authors

While this paper demonstrates strong potential, there may be room for enhancement in certain areas to align it more closely with the publication criteria of this journal. Otherwise, I have a lot of recommendations to increase the quality of your manuscript. Be careful with the writing and mistakes.

This is an interesting article about the plants in the Taihu Lake and its eutrophication, but you must fix a lot of important mistakes in your paper.

There are three keywords repeated in the article title. The keywords are “Taihu Lake”, “herbicide” and “submerged plants”. In order to increase the visibility of your paper I recommend changing these keywords. If you change them by other keywords, you will increase the probability that your paper could be found by future readers when they look for your paper in some databases like Scopus for example. If you repeat the same words in the article title and in keywords, less people could find your work. So, you must think about the visibility of your research.

The redaction of the abstract is very poor, and a very important point is that I cannot identify the conclusions.

Line 29. Just write the economic and cultural services to human society, this will increase the quality of your paper. You must to be much more explicit.

Line 30. A very common mistake throughout your whole article is that you must write a space just before the reference as follows: “…society [1], but are currently…”. This is a very common mistake, and you must fix it. Look for this mistake and I can see that is everywhere in the whole manuscript. Just follow the rules of this journal. You must download a paper of this journal and copy its style, is very easy.

Line 38. You must write “[6,7]” instead of “[6, 7]”. So you must delete the space just before the second reference. When there are two references you must write them together because this is the style of this journal. This is a very common mistake, and you must fix it. Look for this mistake and I can see that is everywhere in the whole manuscript. Just follow the rules of this journal. You must download a paper of this journal and copy its style, is very easy. You must make the work of the editors as easy as possible.

Line 40. I do not understand why you write “zooplankter” instead of “zooplankton”. You must explain this in the text.

Line 72. When you write an author, you must write the reference. So, you must write as follows: “Kemp et al. [the number of the reference] pointed out that exposure…”. This is a very common mistake, and you must fix it. Look for this mistake and I can see that is everywhere in the whole manuscript. Just follow the rules of this journal. You must download a paper of this journal and copy its style, is very easy. You must make the work of the editors as easy as possible.

Line 81. I have found a very important mistake in your manuscript. The sections are not well defined or located, so the sentence: “2. Succession of submerged plants in the Taihu Lake” must be a subsection of the introduction. And in your paper, there is no introduction. Just after the keywords in the line 29 you must write it. This is a very common mistake, and you must fix it. Look for this mistake and I can see that is everywhere in the whole manuscript. Just follow the rules of this journal. You must download a paper of this journal and copy its style, is very easy. You must make the work of the editors as easy as possible.

Line 87. Here you must write the results.

As well you must write the Materials and Methods section just before the Results section.

You must write the objectives of this paper as well, just before the Materials and Methods.

Line 89-91. Rephrase this sentence because is a little bit confusing.

Line 89-95. Just at the left of the text there is no justification. Fix this mistake. You must justify the text in the left of the text.

Line 98. Those manual records must be in a map and a table.

Line 103. You have written the word “and” in italics and this is a mistake because is not a scientific name.

Line 105. “Curly Pondweed” is not a scientific name. Only the scientific names must be written in italics. This is a scientific rule.

Line 101. When you write an author, you must write the reference. So, you must write as follows: “An et al. [the number of the reference] pointed out that exposure…”. This is a very common mistake, and you must fix it. Look for this mistake and I can see that is everywhere in the whole manuscript. Just follow the rules of this journal. You must download a paper of this journal and copy its style, is very easy. You must make the work of the editors as easy as possible.

Line 104. When you write an author, you must write the reference. So, you must write as follows: “Gao et al. [the number of the reference] pointed out that exposure…”. This is a very common mistake, and you must fix it. Look for this mistake and I can see that is everywhere in the whole manuscript. Just follow the rules of this journal. You must download a paper of this journal and copy its style, is very easy. You must make the work of the editors as easy as possible.

Line 107. When you write an author, you must write the reference. So, you must write as follows: “Dong et al. [the number of the reference] pointed out that exposure…”. This is a very common mistake, and you must fix it. Look for this mistake and I can see that is everywhere in the whole manuscript. Just follow the rules of this journal. You must download a paper of this journal and copy its style, is very easy. You must make the work of the editors as easy as possible.

Line 109. You must write a point just after the word “al”.

Line 109. When you write an author, you must write the reference. So, you must write as follows: “Li et al. [the number of the reference] pointed out that exposure…”. This is a very common mistake, and you must fix it. Look for this mistake and I can see that is everywhere in the whole manuscript. Just follow the rules of this journal. You must download a paper of this journal and copy its style, is very easy. You must make the work of the editors as easy as possible.

You must create a table with the species and its families. In this table you must write the authors of the scientific names as well.

Line 122. When you write an author, you must write the reference. So, you must write as follows: “Zhang et al. [the number of the reference] pointed out that exposure…”. This is a very common mistake, and you must fix it. Look for this mistake and I can see that is everywhere in the whole manuscript. Just follow the rules of this journal. You must download a paper of this journal and copy its style, is very easy. You must make the work of the editors as easy as possible.

Line 132-133. I do not understand why you have written these spaces between the paragraphs.

Line 134. When you write an author, you must write the reference. So, you must write as follows: “Wang et al. [the number of the reference] pointed out that exposure…”. This is a very common mistake, and you must fix it. Look for this mistake and I can see that is everywhere in the whole manuscript. Just follow the rules of this journal. You must download a paper of this journal and copy its style, is very easy. You must make the work of the editors as easy as possible.

Line 134. You must write “…for Taihu Lake (Figure 2), and have also reported…” instead of “…for Taihu Lake (Fig. 2), and have also reported…”.

You have forgotten the Figure 2 in this article and even in the Supplementary material there is not this figure.

In this article you must create a map of the location, this is compulsory.

There is no Discussion section.

You must change the all the references because they do not follow the style of this journal.

You must write in bold the year, not the volume of the references. The volume must be written in italics without bold.

Some journals are written in capitals and this is a mistake. You must look for its abbreviation in Journal Citation Reports ®.

You must write the doi of all the references at the very end of each one.

You must write a long hyphen just between the pages of the references.

You must write all the journals in italics.

You must write a comma just after the volume of the reference.

Look for these mistakes and I can see that is everywhere in the whole manuscript. Just follow the rules of this journal. You must download a paper of this journal and copy its style, is very easy. You must make the work of the editors as easy as possible.

There are no Materials and Methods section in this paper, and this is a big mistake of this article. You cannot differentiate the sections of this manuscript.

This article needs major corrections to publish in this journal.

Comments on the Quality of English Language

The English must be improved.

Author Response

There are three keywords repeated in the article title. The keywords are “Taihu Lake”, “herbicide” and “submerged plants”. In order to increase the visibility of your paper I recommend changing these keywords. If you change them by other keywords, you will increase the probability that your paper could be found by future readers when they look for your paper in some databases like Scopus for example. If you repeat the same words in the article title and in keywords, less people could find your work. So, you must think about the visibility of your research.

This is a good question, we modify the keyword to "Taihu Lake basin, shallow lake, herbicide, submerged plants, aquatic environmental security, ecological health.

The redaction of the abstract is very poor, and a very important point is that I cannot identify the conclusions.

Thank you for your question. We have revised the abstract and added statements of relevant conclusions. The details are as follows: According to the results reported in the past paper, the herbicides concentration of environmental in Taihu Lake has sometimes reached a level that can affect a variety of submerged plants, especially in the germination stage, which means that as an important cause of the degradation of submerged plants in shallow lakes, effect of herbicides need to be paid more attention.

Line 29. Just write the economic and cultural services to human society, this will increase the quality of your paper. You must to be much more explicit.

Thank you for your question. In order to avoid duplication with existing conclusions, we have deleted the relevant descriptions of Line 29.

Line 30. A very common mistake throughout your whole article is that you must write a space just before the reference as follows: “…society [1], but are currently…”. This is a very common mistake, and you must fix it. Look for this mistake and I can see that is everywhere in the whole manuscript. Just follow the rules of this journal. You must download a paper of this journal and copy its style, is very easy.

Thank you for your question. We have corrected this error in the full manuscript.

Line 38. You must write “[6,7]” instead of “[6, 7]”. So you must delete the space just before the second reference. When there are two references you must write them together because this is the style of this journal. This is a very common mistake, and you must fix it. Look for this mistake and I can see that is everywhere in the whole manuscript. Just follow the rules of this journal. You must download a paper of this journal and copy its style, is very easy. You must make the work of the editors as easy as possible.

Thank you for your question. We have corrected this error in the full manuscript.

Line 40. I do not understand why you write “zooplankter” instead of “zooplankton”. You must explain this in the text.

Thank you for your question. We have fixed the spelling error.

Line 72. When you write an author, you must write the reference. So, you must write as follows: “Kemp et al. [the number of the reference] pointed out that exposure…”. This is a very common mistake, and you must fix it. Look for this mistake and I can see that is everywhere in the whole manuscript. Just follow the rules of this journal. You must download a paper of this journal and copy its style, is very easy. You must make the work of the editors as easy as possible.

Thank you for your question. We have corrected this error in the full manuscript.

Line 81. I have found a very important mistake in your manuscript. The sections are not well defined or located, so the sentence: “2. Succession of submerged plants in the Taihu Lake” must be a subsection of the introduction. And in your paper, there is no introduction. Just after the keywords in the line 29 you must write it. This is a very common mistake, and you must fix it. Look for this mistake and I can see that is everywhere in the whole manuscript. Just follow the rules of this journal. You must download a paper of this journal and copy its style, is very easy. You must make the work of the editors as easy as possible.

Thank you for your question. We have added relevant expressions in the introduction: However, the community succession of submerged plants in many shallow lakes in the Taihu Basin has occurred violently in recent decades, and submerged plants in some lakes have almost disappeared.

Line 87. Here you must write the results.

Thank you for your question. We have added relevant expressions in Line 87: and our views on whether herbicide may be an important factor in the disappearance or difficult regeneration of submerged plants in Taihu Lake were put forward.

As well you must write the Materials and Methods section just before the Results section.

Unfortunately, as a review type article, this article cannot include the materials and methods sections.

You must write the objectives of this paper as well, just before the Materials and Methods.

Thank you for your question. We have added relevant expressions: The aimed of this review offer guiding for promoting the science-based and standard use of herbicides and preventing their ecological risks in this region.

Line 89-91. Rephrase this sentence because is a little bit confusing.

Thank you for your question. We have revised relevant expressions: Based on the stress effects of herbicides on submerged plants reported in previous studies, whether the submerged plants will be affected by the herbicide concentration in Taihu Lake was analyzed, and our views on whether herbicide may be an important factor in the disappearance or difficult regeneration of submerged plants in Taihu Lake were put forward.

Line 89-95. Just at the left of the text there is no justification. Fix this mistake. You must justify the text in the left of the text.

Thank you for your question. We fixed the formatting error.

Line 98. Those manual records must be in a map and a table.

Thank you for your question. We added Fig. 2.

Line 103. You have written the word “and” in italics and this is a mistake because is not a scientific name.

Thank you for your question. We fixed the formatting error.

Line 105. “Curly Pondweed” is not a scientific name. Only the scientific names must be written in italics. This is a scientific rule.

Thank you for your question. We fixed the formatting error.

Line 101. When you write an author, you must write the reference. So, you must write as follows: “An et al. [the number of the reference] pointed out that exposure…”. This is a very common mistake, and you must fix it. Look for this mistake and I can see that is everywhere in the whole manuscript. Just follow the rules of this journal. You must download a paper of this journal and copy its style, is very easy. You must make the work of the editors as easy as possible.

Thank you for your question. We fixed the formatting error.

Line 104. When you write an author, you must write the reference. So, you must write as follows: “Gao et al. [the number of the reference] pointed out that exposure…”. This is a very common mistake, and you must fix it. Look for this mistake and I can see that is everywhere in the whole manuscript. Just follow the rules of this journal. You must download a paper of this journal and copy its style, is very easy. You must make the work of the editors as easy as possible.

Thank you for your question. We fixed the formatting error.

Line 107. When you write an author, you must write the reference. So, you must write as follows: “Dong et al. [the number of the reference] pointed out that exposure…”. This is a very common mistake, and you must fix it. Look for this mistake and I can see that is everywhere in the whole manuscript. Just follow the rules of this journal. You must download a paper of this journal and copy its style, is very easy. You must make the work of the editors as easy as possible.

Thank you for your question. We fixed the formatting error.

Line 109. You must write a point just after the word “al”.

Thank you for your question. We fixed the formatting error.

Line 109. When you write an author, you must write the reference. So, you must write as follows: “Li et al. [the number of the reference] pointed out that exposure…”. This is a very common mistake, and you must fix it. Look for this mistake and I can see that is everywhere in the whole manuscript. Just follow the rules of this journal. You must download a paper of this journal and copy its style, is very easy. You must make the work of the editors as easy as possible.

Thank you for your question. We fixed the formatting error.

You must create a table with the species and its families. In this table you must write the authors of the scientific names as well.

Unfortunately, we feel that the statement in section 2.1 is sufficient and does not require additional tables.

Line 122. When you write an author, you must write the reference. So, you must write as follows: “Zhang et al. [the number of the reference] pointed out that exposure…”. This is a very common mistake, and you must fix it. Look for this mistake and I can see that is everywhere in the whole manuscript. Just follow the rules of this journal. You must download a paper of this journal and copy its style, is very easy. You must make the work of the editors as easy as possible.

Thank you for your question. We fixed the formatting error.

Line 132-133. I do not understand why you have written these spaces between the paragraphs.

Thank you for your question. We fixed the formatting error.

Line 134. When you write an author, you must write the reference. So, you must write as follows: “Wang et al. [the number of the reference] pointed out that exposure…”. This is a very common mistake, and you must fix it. Look for this mistake and I can see that is everywhere in the whole manuscript. Just follow the rules of this journal. You must download a paper of this journal and copy its style, is very easy. You must make the work of the editors as easy as possible.

Thank you for your question. We fixed the formatting error.

Line 134. You must write “…for Taihu Lake (Figure 2), and have also reported…” instead of “…for Taihu Lake (Fig. 2), and have also reported…”.

Thank you for your question. We fixed the formatting error.

You have forgotten the Figure 2 in this article and even in the Supplementary material there is not this figure.

Thank you for your question. We fixed the formatting error.

In this article you must create a map of the location, this is compulsory.

Thank you for your question. We have added Figure 2.

There is no Discussion section.

Unfortunately, as a review type article, we present our views in each chapter rather than discussing them separately.

You must change the all the references because they do not follow the style of this journal.

You must write in bold the year, not the volume of the references. The volume must be written in italics without bold.

Thank you for your question. We fixed the formatting error.

Some journals are written in capitals and this is a mistake. You must look for its abbreviation in Journal Citation Reports ®.

Thank you for your question. We fixed the formatting error.

You must write the doi of all the references at the very end of each one.

Thank you for your question. We fixed the formatting error.

You must write a long hyphen just between the pages of the references.

Thank you for your question. We fixed the formatting error.

You must write all the journals in italics.

Thank you for your question. We fixed the formatting error.

You must write a comma just after the volume of the reference.

Look for these mistakes and I can see that is everywhere in the whole manuscript. Just follow the rules of this journal. You must download a paper of this journal and copy its style, is very easy. You must make the work of the editors as easy as possible.

There are no Materials and Methods section in this paper, and this is a big mistake of this article. You cannot differentiate the sections of this manuscript.

Unfortunately, as a review type article, we cannot and do not need a description of materials and methods.

Reviewer 2 Report

Comments and Suggestions for Authors

Comments on the Quality of English Language

Well presented and explained.

Author Response

Thank you very much for the reviewer's recognition of us. We will continue to improve this work and carry out the next research based on the practical significance behind this work.

Question: Some mention of the main readership of the paper and the role of policy makers and stakeholders needs to be addressed; The role of farmers and local government needs to be highlighted – in terms 
of alternatives to herbicides and government assistance to phase them out; The relevance to the international community is addressed in the paper but this might be advanced even further in the paper;  International co-operation is needed to address such a common problem with some comparison of different strategies and outcomes.

RE: We have added a description in the discussion part, which is as follows:"In the coming period, to protect the ecological health of the water environment, starting from the recovery of submerged plants, policy managers in the Taihu Lake region may need to make a lot of efforts, which must include stricter control of herbicides discharged into the water environment, and actively collect and track more detailed data on the spatial and temporal distribution of herbicides in the local water environment. Of course, these efforts also need to be supported and coordinated by local agricultural practitioners, and it is important to note that all actions need to be carried out without disrupting agricultural production, which may require closer international cooperation, such as the development of more effective and less toxic herbicides".

Round 2

Reviewer 1 Report

Comments and Suggestions for Authors

While this paper demonstrates strong potential, there may be room for enhancement in certain areas to align it more closely with the publication criteria of this journal. Otherwise, I have a lot of recommendations to increase the quality of your manuscript. Be careful with the writing and mistakes.

This is an interesting article about the plants in the Taihu Lake and its eutrophication, but you must fix a lot of important mistakes in your paper.

I have found that you have improved your manuscript in this current version but still there are a lot of my recommendations that must be done to publish in this journal.

Still there are the same three keywords repeated in the article title. The keywords are “Taihu Lake”, “herbicide” and “submerged plants”. In order to increase the visibility of your paper I recommend changing these keywords. If you change them by other keywords, you will increase the probability that your paper could be found by future readers when they look for your paper in some databases like Scopus for example. If you repeat the same words in the article title and in keywords, less people could find your work. So, you must think about the visibility of your research. You must to fix it in order to publish in this journal.

The redaction of the abstract is very poor, and a very important point is that I cannot identify the conclusions.

Line 29. Just write the economic and cultural services to human society, this will increase the quality of your paper. You must to be much more explicit. Please, fix it.

Line 32. In the previous version of your manuscript you wrote Introduction as a keyword which was a mistake but in this current version is still as a keyword. Please, fix it. Read slowly your paper and fix all the mistakes.

Line 34. Just before the Introduction you must write the number 1. Follow the rules of this journal.

Line 40. You must use the long hyphen between the references. Follow the rules of this journal. As I told you in my previous review you must download a paper of this journal and copy its style. Is very easy.

Still I cannot find the Material and Methods section in this current version, please, fix it.

Still I cannot find the Objectives section. Please, fix it.

Line 95. Just between the references you must avoid the space, so you must write “[27,28]” instead of “[27, 28]”.

Line 114. I do not understand why you write in capitals “Curly Pondweed”.

Line 95. Just between the references you must avoid the space, so you must write “[36,37]” instead of “[36, 37]”. This is a very common mistake in your manuscript. Fix it.

You must summarize the species in a Table showing its frequency, families and abundance.

Line 141. You must improve the map. It is not clear where the Lake is. You must show where you have take the data in the map.

Still there is no Discussion section. Fix it.

Still there is no Results section. Fix it.

Still the references are wrong written. Read again my previous recommendations in my former review.

Still you must change all the references because they do not follow the style of this journal. They have improved but there is still a lot of mistakes.

Please, download a paper of this journal and copy its style.

You must avoid “et al.” in the references, you must write all the authors.

Still you must write a long hyphen just between the pages of the references.

Still you must write the doi of all the references at the very end of each one.

In the abbreviations of the journals you must write a point just after the abbreviation.

You must write a comma just after the year of the reference in all the references, not in several of them.

You must avoid the word “and” before the last author.

You must write the volume in italics.

And there are much more mistakes, please fix them all.

There is no Results section.

There is no Discussion section.

There are no Materials and Methods section in this paper, and this is a big mistake of this article. You cannot differentiate the sections of this manuscript.

Comments on the Quality of English Language

The English is good.

Author Response

While this paper demonstrates strong potential, there may be room for enhancement in certain areas to align it more closely with the publication criteria of this journal. Otherwise, I have a lot of recommendations to increase the quality of your manuscript. Be careful with the writing and mistakes.

This is an interesting article about the plants in the Taihu Lake and its eutrophication, but you must fix a lot of important mistakes in your paper.

I have found that you have improved your manuscript in this current version but still there are a lot of my recommendations that must be done to publish in this journal.

  1. Still there are the same three keywords repeated in the article title. The keywords are “Taihu Lake”, “herbicide” and “submerged plants”. In order to increase the visibility of your paper I recommend changing these keywords. If you change them by other keywords, you will increase the probability that your paper could be found by future readers when they look for your paper in some databases like Scopus for example. If you repeat the same words in the article title and in keywords, less people could find your work. So, you must think about the visibility of your research. You must to fix it in order to publish in this journal.

Re: In the last revision, we added two keywords. About “Taihu Lake”, “herbicide” and “submerged plants”, They are at the heart of this discussion and cannot be replaced.

The redaction of the abstract is very poor, and a very important point is that I cannot identify the conclusions.

Re: We revised the summary last time, and our conclusion is that “In Conclusion, according to the results reported in the past paper, the herbicides concentration of environmental in Taihu Lake has sometimes reached a level that can affect a variety of submerged plants, especially in the germination stage, which means that as an important cause of the degradation of submerged plants in shallow lakes, effect of herbicides need to be paid more attention.”

Line 29. Just write the economic and cultural services to human society, this will increase the quality of your paper. You must to be much more explicit. Please, fix it.

Re: This is the last comment and we have removed the reference.

Line 32. In the previous version of your manuscript you wrote Introduction as a keyword which was a mistake but in this current version is still as a keyword. Please, fix it. Read slowly your paper and fix all the mistakes.

Re: That's a good question. We fixed the bug.

Line 34. Just before the Introduction you must write the number 1. Follow the rules of this journal.

Re: That's a good question. We fixed the bug.

Line 40. You must use the long hyphen between the references. Follow the rules of this journal. As I told you in my previous review you must download a paper of this journal and copy its style. Is very easy.

Re: That's a good question. We fixed the bug in this manuscript.

Still I cannot find the Material and Methods section in this current version, please, fix it.

Re: We must reiterate to the reviewers that as a review type article, we did not conduct relevant experiments, so there is no materials and methods section.

Still I cannot find the Objectives section. Please, fix it.

Re: That's a good question. We set out the objective of this paper in the last paragraph of the introduction. “The aimed of this review offer a possible cause from disappearance of submerged plants in Taihu Lake over the years, and to raise public awareness of the potential ecological impacts of herbicides, promoting the science-based and standard use of herbicides and preventing their ecological risks in this region.”

Line 95. Just between the references you must avoid the space, so you must write “[27,28]” instead of “[27, 28]”.

Re: That's a good question. We fixed the bug in this manuscript.

Line 114. I do not understand why you write in capitals “Curly Pondweed”.

Re: That's a good question. We fixed the bug in this manuscript.

Line 95. Just between the references you must avoid the space, so you must write “[36,37]” instead of “[36, 37]”. This is a very common mistake in your manuscript. Fix it.

Re: That's a good question. We fixed the bug in this manuscript.

You must summarize the species in a Table showing its frequency, families and abundance.

Re: We have added Table 1, which is detailed in line 153 of the manuscript.

Line 141. You must improve the map. It is not clear where the Lake is. You must show where you have take the data in the map.

Re: As a review article, previous researchers have carried out a survey on the distribution of submerged plants in all areas of Taihu Lake. We have divided Taihu Lake into different locations just to make readers more intuitively understand the approximate locations of the place names they read. Our pictures can fully achieve our expectations.

Still there is no Discussion section. Fix it.

Re: We added the results and discussion section. Details are on line 577.Still there is no Results section.

Still there is no Results section. Fix it.

Re: We added the results and discussion section. Details are on line 577.Still there is no Results section.

Still the references are wrong written. Read again my previous recommendations in my former review.

Re: We revised the references as required

Still you must change all the references because they do not follow the style of this journal. They have improved but there is still a lot of mistakes.

Re: We revised the references as required

Please, download a paper of this journal and copy its style.

Re: We revised the references as required

You must avoid “et al.” in the references, you must write all the authors.

Re: We revised the references as required

Still you must write a long hyphen just between the pages of the references.

Re: We revised the references as required

Still you must write the doi of all the references at the very end of each one.

Re: We revised the references as required

In the abbreviations of the journals you must write a point just after the abbreviation.

Re: We revised the references as required

You must write a comma just after the year of the reference in all the references, not in several of them.

Re: We revised the references as required

You must avoid the word “and” before the last author.

Re: We revised the references as required

You must write the volume in italics.

Re: We revised the references as required

And there are much more mistakes, please fix them all.

Re: We revised the references as required

There are no Materials and Methods section in this paper, and this is a big mistake of this article. You cannot differentiate the sections of this manuscript.

Re: We must reiterate to the reviewers that as a “review” type article, we did not conduct relevant experiments, so there is no materials and methods section.

Round 3

Reviewer 1 Report

Comments and Suggestions for Authors

While this paper demonstrates strong potential, there may be room for enhancement in certain areas to align it more closely with the publication criteria of this journal. Otherwise, I have a lot of recommendations to increase the quality of your manuscript. Be careful with the writing and mistakes.

This is an interesting article about the plants in the Taihu Lake and its eutrophication, but you must fix a lot of important mistakes in your paper.

I have found that you have improved your manuscript in this current version but still there are a lot of my recommendations that must be done to publish in this journal. And this is my third revision of this paper. You must follow my recommendations in order to publish in this journal.

I said in my previous reviews that you must download a paper of this journal and copy its style. It is very easy.

Still there are the same three keywords repeated in the article title. The keywords are “Taihu Lake”, “herbicide” and “submerged plants”. In order to increase the visibility of your paper I recommend changing these keywords. If you change them by other keywords, you will increase the probability that your paper could be found by future readers when they look for your paper in some databases like Scopus for example. If you repeat the same words in the article title and in keywords, less people could find your work. So, you must think about the visibility of your research. You must to fix it in order to publish in this journal.

I insist, you must write different keywords to the article title.

One more time you must to write the keywords in alphabetical order.

Other recommendations are the following ones.

Line 35. Which one are the cultural services referred in this line?

Line 116. Curly pondweed is not a scientific name, so you must avoid the italics. You must use the italics only for scientific names.

The very first time that you write a species in the text with its scientific name you must write its authors.

Line 146. You must write frequency in capitals because is the very beginning of a sentence and this is grammatical rule.

In Table 1 you must write the full scientific name and its authors.

Line 149. Just after Wang et al. you must write the reference.

Line 152. You must write the authors of Phragmites australis. The very first time that you write a scientific name you must write its authors. And this is an example.

Line 473. You must write the reference into brackets of Richardson et al.

And I can see that the references do not follow the style of this journal. You must write the name of the authors just after its surname, but only the very first capital letter of the name, not the full name.

I insist, you must download a paper of this journal and copy its style.

And there are much more mistakes, please fix them all. You must read slowly your paper and follow the rules of this journal.

You must use a better statistical tool to publish in this journal, there are no graphics where you can see the frequency of the herbicides at least.

Comments on the Quality of English Language

The English is good.

Author Response

  1. I said in my previous reviews that you must download a paper of this journal and copy its style. It is very easy.

Re: We downloaded the template and carefully modified our manuscript. At the same time, we will be writing to the editor in the hope that we can get the editor's help to further regulate the format issues that we have not considered.

  1. Still there are the same three keywords repeated in the article title. The keywords are “Taihu Lake”, “herbicide” and “submerged plants”. In order to increase the visibility of your paper I recommend changing these keywords. If you change them by other keywords, you will increase the probability that your paper could be found by future readers when they look for your paper in some databases like Scopus for example. If you repeat the same words in the article title and in keywords, less people could find your work. So, you must think about the visibility of your research. You must to fix it in order to publish in this journal. I insist, you must write different keywords to the article title.

Re: We have seriously considered your suggestion and changed the corresponding keyword to “Aquatic environmental security; aquatic vegetation; ecological health1; pesticides; shallow lake”. We believe that today's keywords are broader and more accessible to readers.

  1. One more time you must to write the keywords in alphabetical order

Re: We changed the order of keywords.

  1. Line 35. Which one are the cultural services referred in this line?

Re: We misunderstand the intent of the original document and have deleted this statement.

  1. Line 116. Curly pondweed is not a scientific name, so you must avoid the italics. You must use the italics only for scientific names.

Re: We fixed this formatting error.

  1. The very first time that you write a species in the text with its scientific name you must write its authors.

Re: This is a very good question. Unfortunately, we have consulted relevant references and found that for the use of scientific names for submerged plants, previous researchers have referred to the Flora of China. However, we cannot confirm the identity of the authors who named these submerged plants.

  1. Line 146. You must write frequency in capitals because is the very beginning of a sentence and this is grammatical rule.

Re: We fixed this formatting error.

  1. In Table 1 you must write the full scientific name and its authors.

Re: As with question 6, we cannot confirm the identity of the authors who named these submerged plants.

  1. Line 149. Just after Wang et al. you must write the reference

Re: We corrected this reference error by mistake.

  1. Line 152. You must write the authors of Phragmites australis. The very first time that you write a scientific name you must write its authors. And this is an example.

Re: As with question 6, we cannot confirm the identity of the authors who named these submerged plants.

  1. Line 473. You must write the reference into brackets of Richardson et al.

Re: We fixed this reference error.

  1. And I can see that the references do not follow the style of this journal. You must write the name of the authors just after its surname, but only the very first capital letter of the name, not the full name.

Re: We have corrected this error in full manuscript.

  1. I insist, you must download a paper of this journal and copy its style.

Re: We downloaded the template and carefully modified our manuscript.

  1. And there are much more mistakes, please fix them all. You must read slowly your paper and follow the rules of this journal.

Re: We downloaded the template and carefully modified our manuscript.

  1. You must use a better statistical tool to publish in this journal, there are no graphics where you can see the frequency of the herbicides at least.

Re: This is a good question and we have added Table 4 in the relevant section “Herbicides with high frequency and their types were detected in Taihu Lake area.”
